# Investigating the Blind Spot of a Monitoring System for Article Processing Charges

**Andre Bruns and Niels Taubert ***

Institute for Interdisciplinary Studies of Science (I2SoS), Bielefeld University, Universitätsstraße 25, 33615 Bielefeld, Germany; andre.bruns@uni-bielefeld.de
* Correspondence: niels.taubert@uni-bielefeld.de

**Abstract:** The Open Access (OA) publishing model that is based on article processing charges (APC) is often associated with the potential for more transparency regarding the expenditures for publications. However, the extent to which transparency can be achieved depends not least on the completeness of data in APC monitoring systems. This article investigates two blind spots of the largest collection of APC payment information, OpenAPC. It aims to identify likely APC-liable publications for German universities that contribute to this system and for those that do not provide data to it. The calculation combines data from Web of Science, the ISSN-Gold-OA-list and OpenAPC. The results show that for the group of universities contributing to the monitoring system, more than half of the APC payments are not covered by it and the average payments for non-covered APCs is higher than for APCs covered by the system. In addition, the group of universities that do not contribute to OpenAPC accounts for two thirds of the number of APC-liable publications recorded for contributing universities. Regarding the size of these blind spots, the value of the monitoring system is limited at present.

**Keywords:** open access; article processing charges; monitoring systems



## 1. Introduction

In recent years, a number of activities could be observed that aim to support the transformation towards Gold Open Access (OA), which is based on article processing charges (APC). A number of research institutions created central funds to cover publication fees for OA publications of their authors and established structures and workflows for the organization of payments. On the level of countries, nation-wide OA-contracts have been negotiated, making the bulk of the publication output OA. An important advantage that is often associated with the APC-based OA publishing model is that it has the potential for more transparency regarding the expenditures and financial flows for publications. In the subscription-based model, details of the licenses are usually kept secret as subscription contracts often contain nondisclosure agreements [1,2]. In the APC-based publishing model, the introduction of monitoring instruments has changed this situation. Data collections like OpenAPC [3] include data of actual APC-payments spent by publication funds of research institutions and funding organizations and allow us to deepen our understanding of the OA transformation. However, the value of such monitoring instruments not only depends on the creation of standardized procedures and reporting routines for quality controlled and comparable data but also on the size and completeness of the data they include. An ideal APC monitoring instrument would cover complete APC payments from all research organizations of a given domain. In the real world, APC monitors are lacking in two respects: not all institutions in a given domain deliver data to APC monitors, mostly because of the fact that not all of them have a central publication fund that processes APC payments and collects the data of these transactions [4]. Moreover, research institutions that have a central publication fund are often unable to report all payments of the institution to the monitoring systems as a number of APC payments are made by different entities of a

research organization and are processed in various ways [5]. Such uncovered payments are sometimes referred to as APC 'paid in the wild' [6,7].

This article focuses on the world's largest collection of APC payments, the OpenAPC dataset [3], and addresses both shortfalls for articles in full OA journals (often referred to as 'Gold OA'). Taking German universities as an empirical example, it investigates to what extent APC-liable publications and related payments of universities that report data to OpenAPC are covered in that dataset. In addition, the expenditures for universities that do not contribute to OpenAPC is estimated. The reason for choosing German universities as an example is in part conceptual as well as practical. Regarding conceptual reasons, the German university landscape is a good example since it includes a considerable group of universities that contribute to OpenAPC as well as a number of non-contributing universities, thus allowing an investigation of both blind spots. With respect to practical reasons, the institutional coding that is used for the identification of publications of universities is limited to Germany only.

*Relevance*

The question of the completeness of the OpenAPC monitoring instrument is not only relevant in the context of science policy, which aims to achieve a transformation towards OA and for the management of the process on the local level, but also with regards to the following research topics, which explore the APC-based publishing model.

A first tier of studies starts with the observation that not all full OA journals charge publication fees. Journals that do not come with any financial barriers for readers as well as for authors are called platinum [8,9] or diamond OA [10]. In 2014, roughly two thirds of the journals on the global level refrained from imposing APC [11] and are financed by other means such as subsidies from the state as in the case of Brazil, grants and support from learned societies, or they are driven by the voluntary work of dedicated scientists [12]. The adoption of the APC model seems to differ by field [13,14] and varies by region. A large share of OA journals that do not charge APC can be found in Latin America [12], the Middle East, and Eastern Europe [13].

A second relevant set of studies analyzes the prices for publishing in an APC environment. Due to the lack of other data, early studies referred to list prices on publishers' websites [11] or to prices as recorded by DOAJ [12]. Given that the amount of money that is actually paid for APC may differ from list prices, and given that payments for articles published in the same journal may also vary, more recent studies draw on collections of actual payments since such data collections are now available [3]. Regarding average prices paid for APC, the reported numbers vary between 905 EUR [13] and 1479 EUR [3] with a tendency to increase over time [15]. One peculiarity is that all studies report large standard deviations, indicating that there is much variance in the pricing of APC by the publishers.

The explanation of such difference in pricing is a third topic for which collections of information on APC payments are being used. There is evidence that APC prices are higher for publications in hybrid journals than in full OA journals [16–20] and that they vary by discipline. A study by Solomon & Björk [21] published in 2012 reports higher APC in education, social sciences, law and political sciences as well as in health sciences, biology and life sciences. Besides its age, the relevance of this result, however, is limited due to the small sample of articles. A second analysis by the same authors, also published in 2012, reports larger publication fees for biomedical science and earth science [22]. Another determinant is tested to explain differences of prices for APC: the journal reputation as measured by journal metrics. Budzinski et al. [20] report a positive correlation between the Journal Impact Factor (JIF) and APC. Björk and Solomon [23] as well as Schönfelder [18] use the Source Normalized Impact per Paper (SNIP) as a proxy for journal reputation and come to similar results. This, at least in part, contradicts the findings of Asai [24], who identified two strategies of publishers with respect to APC. The first one is the maximization of revenues of established OA publishers where APC positively correlates with citation rates, the second one where APC positively correlates with the

number of articles, which is interpreted as a strategy to attract more submissions. Finally, the type of publisher is another determinant discussed in the literature. Although the contributions differ regarding the typology of publishers that is used it seems to have an effect. Budzinski et al. [20] find a positive correlation between the size of the publisher and APC and also a positive correlation between the age of a publisher and APC. For medicine, Asai [14] reports significantly higher APC for publications by the five largest publishing houses (Elsevier, SAGE, Springer, Taylor & Francis, and Wiley), while Schönfelder [18] identifies certain publishers as relevant for the explanation of differences in APC.

## 2. Materials and Methods

For the investigation of the blind spots of OpenAPC, the study uses an 'indirect' approach [5] as it aims to identify possible APC-liable publications in full OA journals in bibliometric databases instead of relying on centrally collected data of universities. For the first blind spot, i.e., the incomplete coverage of payments from universities that report data to OpenAPC (in what follows 'OpenAPC universities'), likely APC-liable publications are identified and compared with publications and payment data recorded in OpenAPC. The second blind spot is investigated by the identification of likely APC-liable publications and the estimation of payments for the group of non-reporting universities ('non-OpenAPC universities').

For this purpose, three sources of data are being used:

- *Web of Science (Wos)*: the Web of Science database hosted by the competence centre for bibliometrics is used to determine the publication output for all German universities. Although WoS is not exhaustive, and it is known for a selective coverage and for various biases [25,26], the advantage of this version of the database is that it is enriched with disambiguated institutional addresses for German institutions [27,28]. This allows us to precisely identify the publication output of research institutions in that source. An exhaustive list of German universities was compiled, and all author-address combinations for the document types 'article' and 'review' with at least one address from a German university were retrieved from the database. This information also includes the identifier of the institution, corresponding author information, first author information publication identifier (DOI and WoS-Identifer), article title, publication year, publication type, number of authors and identifiers of the serial (ISSN). Information on whether or not the university contributes to OpenAPC was added. Since the study is interested in an estimation of APC payments, and the institution of the corresponding author is usually supposed to cover the costs, the publications were fully attributed to the university of the corresponding author.

- *ISSN-Gold-OA-list*: Publications in full OA journals were identified for the entire publication output of German universities covered by WoS. The ISSN-Gold-OA-List (in its version 4.0, of 13 July 2020) was used as a source of evidence for publications in full OA journals [29]. It aggregates different lists of full OA journals, including the Directory of Open Access Journals (DOAJ), PubMedCentral (PMC), the Directory of Open Access Scholarly Resources (ROAD) and full OA journals that appear in OpenAPC. After aggregation, the subsection of full OA journals covered both by WoS and the ISSN-Gold-OA-list is manually controlled as to whether or not they offer open access to all content.

- *OpenAPC*: OpenAPC is used as a source for payment data. For the group of the 41 German OpenAPC universities, operationalized as universities that started contributing to OpenAPC at the latest in 2018 and provided data to the monitoring system for the entire year of 2019, payment data were harvested from OpenAPC on 28 August 2020. In addition, OpenAPC was used as a source for an estimation of payments that are not recorded in the system. For each publication identified as a publication in a full OA journal, it was investigated whether or not an APC payment for a publication in the same journal is recorded in OpenAPC. The average APC for articles published in 2019 in the same journal was used as the best estimation for the cost of a likely

APC-liable publication. In cases where OpenAPC does not provide any payment data for a particular journal in 2018 and 2019 but only for older years, the most recent payment was considered as the best estimation. For the group of the non-OpenAPC universities, i.e., the 45 universities that did not contribute with payment information to OpenAPC in the years 2018 to 2019, likely APC-liable publications were identified and payment information was estimated with the same method as described for OpenAPC universities.

## 3. Results

Table 1 reports the number of recorded publications and payments in OpenAPC for the group of OpenAPC universities in the first two columns. 'Likely APC-liable publications' give the number of publications identified by the ISSN-Gold-OA list where no payments are recorded for the particular publication in OpenAPC but where payments for other publications published in the same journal are available. 'Likely payments' reports the estimated overall costs for likely APC-liable publications, and 'Sum costs' is simply the sum of recorded and estimated APC payments for each university. In addition, the ratio of estimated costs of the sum of costs is given (Estim. payment).

**Table 1.** OpenAPC universities, publications covered by OpenAPC and likely APC-liable publications in 2019.

| University | Pub. in OAPC | APC. in OAPC (€) | Likely APC-Liable Pub. | Estim. Payment (€) | Sum Costs (€) | Estim. Payment (%) |
|---|---|---|---|---|---|---|
| TU München | 460 | 655,713 | 355 | 624,054 | 1,279,767 | 48.8 |
| U. Göttingen | 347 | 537,509 | 169 | 306,936 | 844,445 | 36.4 |
| U. Heidelberg | 297 | 468,323 | 517 | 974,481 | 1,442,804 | 67.5 |
| U. Tübingen | 291 | 469,584 | 216 | 414,796 | 884,380 | 46.9 |
| TU Dresden | 265 | 272,230 | 243 | 453,250 | 725,481 | 62.5 |
| KIT | 232 | 329,661 | 161 | 213,433 | 543,094 | 39.3 |
| U.Erlangen/Nürnb. | 222 | 337,001 | 267 | 428,829 | 765,830 | 56.0 |
| U. Leipzig | 218 | 342,628 | 206 | 358,267 | 700,895 | 51.1 |
| U. Duisburg-Essen | 168 | 260,819 | 158 | 266,998 | 527,817 | 50.6 |
| U. Bremen | 148 | 237,189 | 77 | 103,454 | 340,644 | 30.4 |
| U. Regensburg | 134 | 245,730 | 94 | 156,424 | 402,153 | 38.9 |
| U. Bielefeld | 122 | 186,887 | 32 | 47,313 | 234,200 | 20.2 |
| U. Bochum | 113 | 187,325 | 178 | 267,035 | 454,360 | 58.8 |
| FU Berlin | 112 | 157,778 | 209 | 354,521 | 512,299 | 69.2 |
| U. Potsdam | 111 | 167,636 | 82 | 133,477 | 301,113 | 44.3 |
| TiHo Hannover | 108 | 175,247 | 16 | 26,848 | 202,095 | 13.3 |
| U. Münster | 104 | 165,475 | 235 | 391,003 | 556,479 | 70.3 |
| U. Oldenburg | 101 | 156,532 | 46 | 68,422 | 224,955 | 30.4 |
| TU Braunschweig | 101 | 121,605 | 73 | 99,766 | 221,370 | 45.1 |
| U. Rostock | 99 | 134,823 | 105 | 152,671 | 287,493 | 53.1 |
| U. Mainz | 95 | 152,970 | 249 | 375,772 | 528,742 | 71.1 |
| U. Hannover | 91 | 138,968 | 82 | 111,480 | 250,448 | 44.5 |
| LMU München | 89 | 158,864 | 559 | 985,012 | 1,143,876 | 86.1 |
| TU Berlin | 86 | 123,275 | 102 | 147,500 | 270,774 | 54.5 |
| U. Gießen | 75 | 119,171 | 204 | 331,799 | 450,970 | 73.6 |
| U. Halle-Wittenb- | 73 | 116,371 | 94 | 152,925 | 269,296 | 56.8 |
| TU Darmstadt | 73 | 106,245 | 85 | 100,957 | 207,202 | 48.7 |
| U. Bayreuth | 68 | 94,062 | 45 | 62,404 | 156,466 | 39.9 |
| U. Konstanz | 62 | 101,493 | 66 | 99,849 | 201,342 | 49.6 |
| U. Kassel | 62 | 81,087 | 16 | 24,557 | 105,644 | 23.3 |
| U. Stuttgart | 51 | 67,423 | 87 | 129,042 | 196,465 | 65.7 |
| TU Dortmund | 38 | 47,619 | 57 | 71,238 | 118,857 | 59.9 |
| U. Osnabrück | 37 | 59,296 | 31 | 51,383 | 110,680 | 46.4 |
| TU Chemnitz | 29 | 36,794 | 43 | 53,903 | 90,697 | 59.4 |
| TU Hamb.Harburg | 22 | 31,469 | 17 | 19,247 | 50,715 | 38.0 |
| TUIlmenau | 21 | 29,560 | 21 | 18,360 | 47,920 | 38.3 |
| U. Bamberg | 19 | 31,180 | 3 | 3604 | 34,784 | 10.4 |
| TU Clausthal | 13 | 17,825 | 15 | 15,369 | 33,194 | 46.3 |
| U. Siegen | 9 | 11,298 | 29 | 35,076 | 46,375 | 75.6 |
| U. Mannheim | 9 | 15,880 | 15 | 12,486 | 28,366 | 44.0 |
| U. Passau | 1 | 829 | 8 | 6732 | 7560 | 89.0 |
| Total | 4776 | 7,151,375 | 5267 | 8,650,673 | 15,802,048 | 54.7 |

The results reveal that in 2019 payment data from universities reported to OpenAPC are far from being complete. In total, more than half of the costs are not covered by OpenAPC in the group of OpenAPC universities. Examples of universities with a large publication output and large non-covered parts are LMU München, Universität Heidelberg, FU Berlin, Universität Münster, and TU Dresden. In the case of the LMU München, more than 85% of the estimated payments are not included. On the other side of the spectrum,

nearly complete coverage is rare. Universität Bamberg (10.4% estimated payments not covered by OpenAPC) and TiHo Hannover (13.3%) are examples here.

In the interpretation of the results one has to consider that the number of recorded payments in OpenAPC and the estimated number of publications are based on different entities. OpenAPC collects all known payments of a university in its publication output, while the estimation of likely APC-liable publications only considers those parts of the publication output that are covered by the Web of Science database. The remaining parts of a universities' publication output not covered by WoS may also contain a certain fraction of APC-liable publications. Therefore, the ratio and a confidence interval of non-WoS-covered publications of all publications recorded with payment in OpenAPC was calculated for the group of OpenAPC universities. The calculated ratio is 0.146 with a confidence interval with a lower bound of 0.122 and an upper bound of 0.170. When taking the proportions as a correction factor into account, the likely not-covered part is even bigger than reported in Table 1. According to this estimation, 55.8% of the publications and 58.1% of the costs for APC in total are not covered in OpenAPC. These numbers are much higher than the 20% of APC 'paid in the wild' as reported by Andrew [6].

The reasons for the incompleteness of the payment information reported by universities are beyond the scope of this study. Nevertheless, the results seem to be in line with an author survey conducted on behalf of Springer Nature. It reports that publication funds are only one of a number of different sources, including research funders, institutions, and publisher agreements, that are often combined to cover costs for APC [7]. Moreover, publication funds may also not be targeted as a source for APC payments as they are unknown to authors. Thus, authors may use easier accessible funds to cover the cost, or the APC prices may exceed possible limits of publication funds, such as, for example, the price cap of 2000 EUR in the program 'Open Access Publizieren' of Deutsche Forschungs-gemeinschaft (DFG) [30]. In addition, the publication funds of a university of a certain year may have already been spent at the time of the payment of APC [31].

The results of the investigation of the second blind spot of the monitoring system OpenAPC can be found in Table 2. The identification of possible APC-liable publications and the estimation of the related costs follow the same procedure as for the group of OpenAPC universities. Again, the proportion of APC-liable publications in and outside WoS for the group of OpenAPC universities would suggest that the overall number is approximately 12–17% larger.

**Table 2.** Non-OpenAPC universities, APC-liable publications in 2019.

| University | Likely APC-Liable Pub. | Est. Payments (€) |
|---|---|---|
| U. Hamburg | 443 | 803,314 |
| RWTH Aachen | 422 | 741,787 |
| U. Bonn | 302 | 496,702 |
| MHH Hannover | 263 | 521,071 |
| U. Köln | 248 | 487,592 |
| U. Düsseldorf | 246 | 470,823 |
| U. Jena | 232 | 425,050 |
| U. Kiel | 212 | 369,026 |
| U. Magdeburg | 136 | 233,137 |
| U. Hohenheim | 132 | 180,114 |
| U. Lübeck | 108 | 202,482 |
| UK Schleswig-Holstein | 81 | 149,138 |
| TU Kaiserslautern | 62 | 113,612 |
| U. Witten/Herdecke | 61 | 114,421 |
| UK Gießen und Marburg | 44 | 87,147 |
| SHS Köln | 41 | 75,520 |
| U. Wuppertal | 31 | 54,118 |
| TU Bergakademie Freiberg | 29 | 50,023 |
| U. Augsburg | 29 | 43,573 |

**Table 2.** *Cont.*

| University | Likely APC-Liable Pub. | Est. Payments (€) |
|---|---|---|
| Jacobs University Bremen | 25 | 37,530 |
| U. Paderborn | 24 | 40,959 |
| U. Koblenz-Landau | 20 | 34,901 |
| U. Lüneburg | 19 | 30,985 |
| TU Cottbus-Senftenberg | 15 | 16,398 |
| U. Eichstät -Ingolstadt | 11 | 20,220 |
| U. Weimar | 8 | 10,378 |
| U. der BW München | 6 | 12,152 |
| U. Hildesheim | 6 | 9388 |
| Herzzentrum Freiburg | 6 | 10,202 |
| U. der BW Hamburg | 5 | 7716 |
| FernU. Hagen | 4 | 5976 |
| U. Vechta | 4 | 6392 |
| HS für Musik Hannover | 3 | 5469 |
| U. Erfurt | 2 | 4041 |
| MHS Brandenburg | 2 | 3048 |
| PH Freiburg | 2 | 3002 |
| Otto Beisheim School of Mana. | 2 | 1730 |
| Comprehensive Cancer Center | 1 | 2092 |
| PH Karlsruhe | 1 | 1747 |
| PH Schwäbisch Gmünd | 1 | 1746 |
| ESCP Berlin | 1 | 1171 |
| PH Heidelberg | 1 | 1344 |
| Hertie School of Governance | 1 | 3128 |
| HafenCity Universität Hamburg | 1 | 1088 |
| Zeppelin U. | 1 | 2020 |
| Total | 3294 | 5,893,470 |

## 4. Discussion

Our investigation of the blind spots of the monitoring instrument has some limitations.

- First, it aims to indirectly identify likely liable publications in the two groups of universities, by identifying other publications in the same journal where APC have been paid for. Given that journals may change their business model (flip to APC as well as reverse flip to subscription [32]), there might be cases where publications are falsely classified as APC-liable.

- Second, there might be APC-liable publications in journals within the two blind spots of the monitoring system, where no payments for other publications in the same journal were recorded in OpenAPC. Although the monitoring system provides APC payment information for 2411 full OA journals, there might be APC journals that are not covered in this data base. For this reason, the number of likely APC-liable publications might be too small. As a result of the first two limitations, it cannot be decided whether the identified number of APC-liable publications is more a minimum or a maximum estimation for the actual number of APC-liable publications.

- Third, the estimation of costs for likely APC-liable publications is based on payments for other publications in the same journal, for which one or more payments have been recorded. Given that payments may differ, for example, because of changes in the pricing for APC [24], or discounts and waivers included in contracts between publishers and universities, the estimated APC-prices are approximations.

- Fourth, the identification of possible liable publications was performed on a certain part of the publication output of a university which is covered by the Web of Science. The calculation of a correction factor therefore has to be regarded as a rough approximation for the volume of possibly not covered APC-liable publications outside the scope of the WoS database.

- Fifth, the design of the study only allows us to estimate APC-liable publications of a certain OA type, publications in full OA journals, but not other types like publications in hybrid journals. Moreover, it does not take so-called transformative agreements into account that are negotiated by different initiatives in a number of European countries [33]. Given that it is likely that transformative agreements will not be negotiated with all publishing houses, and given that not all countries may follow the path of such OA transformation, the results of the study will remain relevant in the future.

After having described the limitations of the approach, the results shall be discussed in the context of three aspects: the monitoring of the OA transformation, calculation of average OA prices, and the determinants of APC.

*Monitoring of the OA Transformation*: the transformation towards OA comes along with the hope for more transparency regarding financial flows on the publication market. Regarding this hope, the results of this study point to the fact that the incompleteness of payment data in monitoring systems lead to an insufficient information basis for local planning or for a comprehensive (country-wide) OA strategy. Both the number of APC-liable publications and the volume of money are underestimated if only APC-liable publications recorded in systems like OpenAPC are considered.

*Average OA prices:* regarding the calculation of average APC payments, incompleteness of data are not an issue if the recorded APC payments would be an adequate representation of all recorded and non-recorded APC payments. Unfortunately, this does not seem to be the case, at least for German universities. For the group of OpenAPC universities, an average amount of 1497 EUR was calculated for payments covered by OpenAPC compared to an estimated average of 1642 EUR for likely APC-liable publications not covered by OpenAPC. The results of the group of non-OpenAPC universities point in the same direction. The estimated non-covered payments of 1789 EUR are considerably higher than the average payments recorded in OpenAPC. This difference may be caused by price caps of publication funds or by better contracts that are negotiated between libraries and publishers. However, the use of OpenAPC data for the calculation of average APC payments may lead to an underestimation of costs.

*Determinants of APC:* finally, studies that are interested in determinants of OpenAPC may also be affected by a biased selection of APC payments that are included in databases like OpenAPC. Given that higher average prices are correlated with specific commercial publishers [18] or with the age of a publishing house [20], and given that non-recorded payments tend to be higher than those recorded in OpenAPC, the determinant 'type of publisher' might actually explain more of the variance of APC prices than recent studies suggest.

## 5. Conclusions

The result of this study illustrates that, at this stage, the incompleteness of data reported to OpenAPC restricts the value of the monitoring system. For German universities that contribute to OpenAPC, more than half of the estimated expenditures for APC are not recorded in the system. In addition, the number of likely APC-liable publications of German universities that do not contribute to OpenAPC corresponds to 68.9% of the recorded publications. However, these results should not be understood as a critique of APC monitoring systems in general or OpenAPC in particular. In order to unfold their full potential for transparency, the reporting procedures of research organizations need to improve. Capturing APC payments more exhaustively by extracting them from the accounting systems of the financial administration could be one way to go [34,35]. The results of the study, however, do not only call for an improvement of the data in monitoring systems but also point to the need of follow-up studies:

Regarding the publications in full OA journals, the article only gauges the size of the blind spots but does not offer any explanation about the reasons for missing payment data especially in cases of universities that contribute to OpenAPC. For a deeper understanding of APC-related financial flows in universities, interviews with OA representatives might

offer a more in-depth picture. With respect to the decision logic of corresponding authors, especially those at universities that have publication funds but decide to finance APC by other means, an interview study with them might also offer interesting insights.

The starting point of the article was the question whether or not more cost transparency, which is often associated with OA, becomes a reality. However, the study is limited to one specific OA business model, articles in full OA journals financed by APC. For other business models, this question has to be answered separately. First and most important, there are a growing number of transformative agreements between publishers and research organizations in several countries, and it would be interesting to study to what extent and under what conditions such contracts contribute to more cost transparency. Second, there are a large number of diamond OA journals worldwide that run their operation without charging APC. Given that these journals sometimes receive subsidies or are financed in an indirect way, it would also be interesting to see how they perform in financial terms against APC-based journals.

**Author Contributions:** Conceptualization, N.T.; methodology, N.T. and A.B.; formal analysis, A.B.; writing—original draft preparation, N.T.; writing—review and editing, N.T. and A.B.; project administration, N.T.; funding acquisition, N.T. All authors have read and agreed to the published version of the manuscript.

**Funding:** This research was funded by the Federal Ministry of Education and Research (BMBF) grant number 160A32). The APC was funded by Bielefeld University.

**Institutional Review Board Statement:** Not Applicable.

**Informed Consent Statement:** Not Applicable.

**Data Availability Statement:** The data can be requested from the corresponding author on request.

**Acknowledgments:** We are very grateful for the valuable suggestions and comments of four anonymous reviewers.

**Conflicts of Interest:** The authors declare no conflict of interest.

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
