# Peer review of "Investigating the Blind Spot of a Monitoring System for Article Processing Charges"

_publications, doi:10.3390/publications9030041_

Round 1

Reviewer 1 Report

The article raises a crucial question about the transparency of the spenditure on APC publication by country’s governments. The overall conception is very interesting, but the literature review, the data and the discussions is missing a few essential points:

a) The Bigger Deals or Read and Publish agreements as a strategy for de big publishers to negotiate direct with the universities without considering the librarians.

b) The plan S, which not consider hybrid journals as open access

c) How the institutions will contabilizar the expenditure with access and scientific publications with the new models?

p.1 l. 38 - implies Germany does not have a “standardized procedures and reposting routines for quality controlled and comparable data”? The Open APC mentioned does not look updated and seems exclusive for Germany. How about an international tool? For every country?

p.2 l.61 - refers to Gold Open Access or are considering Hybrid not Open Access?

p.2 l. 71 – needs a citation here.

p.2 l. 72 – you cannot say the work is unpaid. It’s part of the salary from their employer, with recognizes the activity as editor in the payable activities.

p.2 l. 87 – hybrid its not Open, see the plan S.

p.2 l.88 – references 22 and 23 are too old (2012) for this analysis.

p.2 l. 71-74 – the journals in open access mentioned here are indexed in WoS? or SCOPUS? Otherwise it’s not comparable

p.3 l. 12 – about the data from WoS did you select all types of access? Even hybrid? Or green or bronze?

p.3 l. 13 – it´s not clear how the Gold OA List works, its just for German universities? Why the table one does not presents the total of publications plus the OA percent? This data will help to estimated the size of AO in Germany.

p.3 l.14 – this works just for German’s Universities? It’s not clear from a not European country how it works.

p.5 and 6 – Why the columns of the two tables are not the same? It’s an estimative? As is its impossible to compare, and the main conclusion is the data collection is not standardized and does not allow usable conclusions.

p.7 l.22 – The second part affirms “it cannot be decided whether the identified number of APC liable publications is more a minimum or a maximum estimation for the APC-liable publications” admits the contribution of the method (and the article) is very restricted.

p.7 l.24 – Hybrid its not Open Access, see plan S. Why does not study the so-called transformative agreements? This strategy demands much studies… German have transformative studies?

p.7 l. 26 – This sentence its not clear, it douts the data of the article?.

Author Response

Dear reviewer,

We would like to thank you for reading our manuscript and for your comments. We feel that some of your comments are very helpful for being more clear in the use of terminology, while others point to different expectations regarding the aim of the study. We therefore comment directly on the points you raised,

The article raises a crucial question about the transparency of the spenditure on APC publication by country’s governments. The overall conception is very interesting, but the literature review, the data and the discussions is missing a few essential points:

  1. The Bigger Deals or Read and Publish agreements as a strategy for de big publishers to negotiate direct with the universities without considering the librarians.

Answer:

The effects of OA business models on cost transparency differ and, from our point of view, have to be studied separately. This study focusses exclusively on the APC-based business models that are processed via publication funds, and cost information are aggregated by a monitoring system.  The situation is different from - for example – the case of PAR agreements where publication and payment information are reported to a single point (in the case of Germany: the MPDL services GmbH). But PAR agreements as well as scholarly led OA journals are not the focus of our paper. However, we included a conclusion section in which we address this as questions where further research is needed.

  1. The plan S, which not consider hybrid journals as open access

Answer:

The question whether or not hybrid journals are OA depends on your definition of OA. In the paper, we are only interested in publications in full OA journals. The term hybrid is used on two occasions. To avoid misunderstanding, we consistently used the term hybrid journals.

  1. How the institutions will contabilizar the expenditure with access and scientific publications with the new models?

Answer:

We would agree that the question as to how research institutions shift their payments from subscription to APC (and what kind of compensation mechanisms between institutions are required) is one of the most serious problems of the APC business model. However, the question is beyond the scope of our manuscript.

p.1 l. 38 - implies Germany does not have a “standardized procedures and reposting routines for quality controlled and comparable data”? The Open APC mentioned does not look updated and seems exclusive for Germany. How about an international tool? For every country?

Answer:

The quotation aims to introduce the perspective that is developed in the paper: not to simply trust that the introduction of an APC monitoring system leads to transparency but to study empirically to what extent transparency can be achieved by such systems in practice.

OpenAPC is indeed an international tool comprising data from 18 countries, including UK, France, US, and many other European countries. It is updated regularly.

p.2 l.61 - refers to Gold Open Access or are considering Hybrid not Open Access?

Answer:

Our analysis refers to articles in full OA journals (i.e. journals making all content open access) only and not to hybrid OA. We avoid the term ‘Gold OA’ because of its ambiguous meaning. Sometimes it is used in the sense of ‘journal based OA’ and sometimes the meaning is restricted to full OA. We therefore feel that full OA is a term that helps to avoid such confusion. We included a short explanation in the text that full OA is sometimes also called ‘Gold’.

p.2 l. 71 – needs a citation here.

Answer:

True. We included a citation.

p.2 l. 72 – you cannot say the work is unpaid. It’s part of the salary from their employer, with recognizes the activity as editor in the payable activities.

Answer:

True. We removed the ‘unpaid’. The work is voluntary in many if not most cases.

p.2 l. 87 – hybrid its not Open, see the plan S.

Answer:

We consistently use the term hybrid journals (instead of hybrid OA) throughout the paper to avoid confusion.

p.2 l.88 – references 22 and 23 are too old (2012) for this analysis.

Answer:

We agree, both references are old. To our knowledge they are the first that found disciplinary differences for the pricing of APC and should therefore be referenced. Our changes now make clear that these studies reflect a past situation.

p.2 l. 71-74 – the journals in open access mentioned here are indexed in WoS? or SCOPUS? Otherwise it’s not comparable

Answer:

All of the three studies referenced on p. 2, l.71-74 are based on the DOAJ. [14] is restricted to a subsection of DOAJ, the medical journals. We do not intend to claim that the results of the studies can be compared directly, as they were conducted at different points in time.

p.3 l. 12 – about the data from WoS did you select all types of access? Even hybrid? Or green or bronze?

Answer:

We selected all publication with an address of a German university from WoS as explained in p3, l.120-135 no matter if they are OA or not (and what type of OA). Our full OA evidence source is the ISSN-Gold-OA list, the APC evidence source is OpenAPC.

p.3 l. 13 – it´s not clear how the Gold OA List works, its just for German universities? Why the table one does not presents the total of publications plus the OA percent? This data will help to estimated the size of AO in Germany.

Answer:

As described (p. 3 l.139), the ISSN-Gold-OA list aggregates different evidence sources for full OA journals (DOAJ, PMC, ROAD and full OA journals that appear in OpenAPC). Therefore, it can be used to identify publications in full OA journals in any set of publications, not only German publications.

The paper does not aim to determine the OA share of German institutions. This is a topic of another paper we published together with colleagues: Hobert, A., Jahn, N., Mayr, P., Schmidt, B., Taubert, N.. Open access uptake in Germany 2010–2018: adoption in a diverse research landscape. Scientometrics (2021). https://doi.org/10.1007/s11192-021-04002-0

p.3 l.14 – this works just for German’s Universities? It’s not clear from a not European country how it works.

Answer:

No, the usage of the ISSN-Gold-OA list is by no means restricted to the German context.

p.5 and 6 – Why the columns of the two tables are not the same? It’s an estimative? As is its impossible to compare, and the main conclusion is the data collection is not standardized and does not allow usable conclusions.

Answer:

Table 1 refers to the group of universities that provide data for OpenAPC. For this group of universities, we have information about APC-liable publications and costs and we can also identify likely APC-liable publication. Table 2 refers to the group of universities that do not provide any data to OpenAPC. For that reason, we do not have any observed numbers (publications and costs) for these institutions but only the number of likely APC-liable publications and the journal-specific estimation of costs.

p.7 l.22 – The second part affirms “it cannot be decided whether the identified number of APC liable publications is more a minimum or a maximum estimation for the APC-liable publications” admits the contribution of the method (and the article) is very restricted.

Answer:

This is true but we believe this is the best possible estimation, based on the collection of data we have today.

p.7 l.24 – Hybrid its not Open Access, see plan S. Why does not study the so-called transformative agreements? This strategy demands much studies… German have transformative studies?

Answer:

We consistently use the term hybrid journals throughout the paper to avoid confusion. 

Regarding the financial effects of DEAL, there is a lack of data up to now and OpenAPC only recently started the implementation of the data of the contracts. To our knowledge there is only a cost-modeling tool available on the Website of MPDL (https://deal-operations.de/das-ist-der-deal/deal-kostenmodellierungstool) and a publication that describes the service (German only, Schimmer, R., Dér, Á., Campbell, C. (2021): Das DEAL Kostenmodellierungstool: Ein praktischer Beitrag zur Bewertung von Wirkung und Kosten hinter transformativen Verlagsverträgen. doi: 10.17617/2.3331716, S. 10 f.

p.7 l. 26 – This sentence its not clear, it douts the data of the article?

Answer:

We revised the sentence to make it more clear.

Again, we would like to thank you for your time and effort you spent on our manuscript.

Best regards

Andre Bruns

Niels Taubert

Reviewer 2 Report

This paper is very innovative. The aim and contributions of the research are clearly stated.

Author Response

We would like to thank you for reading our manuscript and for your encouraging comment!

Reviewer 3 Report

The article is well structured and is clear about the methodology - including its limitations.

Author Response

(The authors gave the same response as above.)

Reviewer 4 Report

I found this to be a clearly outlined article, articulating the issue and providing a thorough methodology. I found the selection of the case study to be an intriguing one, and also the investigation into the two noted blind spots. I think this is an important paper attempting to address the lack of transparency in APCs, and I applaud the efforts of the authors.

I also appreciated the descriptions of each of the limitations of the study- I think this is important to address, and perhaps also some ideas for future studies to address one or more of these limitations.

I would recommend a conclusion section after the discussion section. Something to summarize the article but also address some future avenues of study.

Some minor grammatical errors- Line 51- desiderata- Was this supposed to be "a desire" instead?

I wonder as a follow-up study or project, if individuals from universities working on OA charges (likely the library or research offices) could be interviewed to find out a bit more perspective? (Speaking from my institution, the ways OA charges are covered are variable, and I had not thought of what an interesting piece this could be in the grand scheme in how universities fund and consider OA charges. We have two ways for researchers to fund OA charges, though from my POV these services are not well known to most at our university). And perhaps also a study from the author perspective on the APCs? I think it would be interesting to see what the behavior in journal selection would be.

Author Response

Dear reviewer,

We would like to thank you very much for your comments and suggestions that are all very constructive and helpful. We will respond directly into the points you raised,

Comments and Suggestions for Authors

I also appreciated the descriptions of each of the limitations of the study- I think this is important to address, and perhaps also some ideas for future studies to address one or more of these limitations.

I would recommend a conclusion section after the discussion section. Something to summarize the article but also address some future avenues of study.

Answer:

We included a conclusion section We put together some of the suggestions below with our ideas about possible next steps.

Some minor grammatical errors- Line 51- desiderata- Was this supposed to be "a desire" instead?

Answer:

Thank you for pointing us to that line. We revised the sentence.

I wonder as a follow-up study or project, if individuals from universities working on OA charges (likely the library or research offices) could be interviewed to find out a bit more perspective? (Speaking from my institution, the ways OA charges are covered are variable, and I had not thought of what an interesting piece this could be in the grand scheme in how universities fund and consider OA charges. We have two ways for researchers to fund OA charges, though from my POV these services are not well known to most at our university). And perhaps also a study from the author perspective on the APCs? I think it would be interesting to see what the behavior in journal selection would be.

Answer:

We included this suggestion in the conclusion. Regarding cost transparency in the case of diamond OA journals (mentioned in the conclusion), we just started a project that aims to gather data for an estimation of the costs of such journals (10 cases).

Again, thank you very much for your careful reading, for your encouraging comments and for your suggestions!

Best regards

Andre Bruns

Niels Taubert

Round 2

Reviewer 1 Report

The authors made specific adjustments to the article and ignored the options they considered out of the aim of the study.
My initial recommendation was for rejection, and it still is. The work is confused and based on a hard-to-understand local scenario.